# Generation, transmission, and conversion of orbital torque by an antiferromagnetic insulator

Shilei Ding [1] ✉, Paul Noël [1], Gunasheel Kauwtilyaa Krishnaswamy [1], Niccolò Davitti[1], Giacomo Sala [1], Marzia Fantauzzi[2], Antonella Rossi[1,2] & Pietro Gambardella [1] ✉

Electrical control of magnetization in nanoscale devices can be significantly improved through the efficient generation of orbital currents and their conversion into spin currents. In nonmagnetic/ferromagnetic bilayers, this conversion produces a torque on the magnetization, enabling magnetization switching and dynamic manipulation. While previous studies focus on metallic ferromagnets, we demonstrate a large orbital torque and enhanced orbital-to-spin conversion by an antiferromagnetic insulating CoO layer. Measurements in $CuO_x/CoO/Co$ trilayers show that inserting CoO reverses the torque's sign and triples its magnitude compared to $CuO_x/Co$. This behaviour stems from the inverted oxygen gradient at the $CuO_x/CoO$ interface and CoO's high orbital multiplicity, which favours the transmission of orbital momenta and efficient orbital-to-spin conversion. At low temperatures, the onset of antiferromagnetic order induces a further many-fold increase of the torque, which we attribute to the efficient excitation and propagation of spin-orbit excitons induced by magnetic coupling. Comparative measurements of $CuO_x/NiO/Co$ and $CuO_x/MnO/Co$ trilayers show that the torque efficiency scales with the orbital momentum of the $Co^{2+}$, $Ni^{2+}$, and $Mn^{2+}$ ions in the antiferromagnet. These results reveal that antiferromagnetic insulators like CoO provide highly effective orbital-to-spin transduction, combining orbital torque and exchange bias functionalities to improve the performance of spintronic devices.

The recent discovery of orbital torques in transition-metal systems provides a new mechanism for current-induced manipulation of the magnetization, which is potentially more efficient than spin torques and complementary to them[1–16]. Similar to spin torques, orbital torques can be used for magnetization switching, domain wall motion, and excitation of high-frequency dynamics in nano-oscillators[17]. These phenomena find application in complementary metal-oxide semiconductor-compatible memory and logic architectures[18–20]. Additionally, orbital torques may be used to convey electrical and lattice excitations to magnonic systems[21],

magneto-electric devices[22], and other unconventional computing platforms[23].

Theoretical models indicate that orbital torques arise from the interplay of two fundamental processes: the generation of orbital currents and their conversion into spin torques mediated by spin-orbit coupling (SOC)[1,2]. While orbital magnetic moments are typically quenched in solids, applying an electric field to a nonmagnetic conductor can create a nonequilibrium accumulation of orbital momenta by driving the hybridization of electronic states with distinct orbital characters[2]. Notable examples of this process include the orbital Hall

[1]Department of Materials, ETH Zürich, Zürich, Switzerland. [2]Dipartimento di Scienze Chimiche e Geologiche, Campus di Monserrato, Università degli Studi di Cagliari, Cagliari, Italy. ✉e-mail: shilei.ding@mat.ethz.ch; pietro.gambardella@mat.ethz.ch

effect[24-27] and orbital Rashba-Edelstein effect[28-30], which are known to occur in transition metals[11-15,31,32], 2D materials[33-35], and oxide interfaces[3-10], respectively, even in the absence of SOC. The diffusion of nonequilibrium orbital momenta into an adjacent magnetic material induces an orbital current that is absorbed by the local magnetization, generating an orbital torque. This process, instead, relies heavily on SOC to mediate the interaction between the orbital current and the spin magnetization. Consequently, orbital torques are usually prominent in ferromagnets with strong SOC, such as Ni[11-14], ferrimagnets including rare-earth atoms, such as GdCo[10], and NM/X/FM trilayers where NM is a light-metal orbital source, FM a ferromagnet, and X a spacer layer with large SOC such as Pt, Gd, or Tb[3,12,13,16]. Spacer layers induce orbital-to-spin conversion prior to injection in the ferromagnet and are thus very effective in enhancing the magnitude of the orbital torque. However, most studies on orbital-to-spin conversion processes have focused on metal spacers with high SOC, leaving considerable potential for further progress in this area.

In this work, we investigate insulating antiferromagnets (iAFM) as potential orbital-to-spin transducers. Previous studies have shown that iAFM spacers such as NiO and CoO enable the efficient transfer of spin currents generated by spin pumping[36-41], thermal gradients[42-45], and the spin Hall effect of heavy metal layers[46,47]. These effects have been attributed to spin transport mediated by either incoherent thermal magnons and spin fluctuations[36-40,45-51], coherent antiferromagnetic magnons[41-44,52-54], or interfacial exchange bias[46,49,55]. However, it is still an open question whether iAFMs can efficiently transport and convert orbital currents between NM and FM layers. To answer this question, we study the orbital torques induced by current injection in Cu*/CoO/Co and Cu/CoO/Co trilayers as a function of CoO thickness and temperature in comparison to Cu*/Co bilayers. Here, Cu* stands for naturally oxidized Cu, a known source of orbital polarization due to the orbital Rashba-Edelstein effect arising from the hybridization between the oxygen $p$-orbitals and Cu $d$-orbitals[3-10,30]. The choice of CoO as iAFM is motivated by the partially unquenched orbital momentum of the Co$^{2+}$ ions in a cubic crystal field[56], and the consequent mixed spin-orbital character of its electronic and magnonic excitations[57-61]. We further compare the magnitude of the current-induced torques in Cu*/NiO/Co and Cu*/MnO/Co trilayers, showing

that the orbital torque efficiency scales with the orbital moment of the iAFM.

Our findings demonstrate that CoO greatly enhances the generation, transport, and conversion of orbital momenta between Cu and Co, resulting in a many-fold increase in the orbital torque efficiency. The insertion of CoO reverses the oxygen gradient relative to the Cu*/Co bilayer and induces a strong orbital Rashba-Edelstein effect at the Cu/CoO interface. Additionally, the observation of a large orbital torque in the Co layer indicates efficient transmission and conversion of orbital momenta through CoO. Above the antiferromagnetic ordering temperature, these processes are mediated by thermal spin fluctuations. Below the ordering temperature, additional mechanisms emerge, supported by antiferromagnetic exchange coupling, interfacial exchange bias, and orbital magnetic moments of CoO. From a practical standpoint, our work establishes that iAFMs can be used to engineer and amplify orbital torque effects, providing a versatile approach for optimizing spintronic devices.

## Results
### Orbital torque efficiency
Our samples are Cu*($t_{Cu}$)/Co(2) and Cu*(7)/CoO($t_{CoO}$)/Co(3) heterostructures (layer thickness in nanometers) deposited on SiO₂/Si substrates by magnetron sputtering, as schematically shown in Fig. 1a, b. Additional samples with unoxidized Cu and without Cu were prepared for control purposes. To obtain an oxygen gradient in Cu*, we let the Cu naturally oxidize in the air for 2 days before starting the measurements[3], whereas CoO was grown by either post-deposition oxidation of Co in a vacuum or direct deposition from a stoichiometric CoO target (see Methods). The root mean square surface roughness measured by atomic force microscopy is about 0.4 nm (Supplementary Fig. S1). Elemental depth profiling by X-ray photoelectron spectroscopy (XPS) shows that natural oxidation of Cu leads to the formation of Cu₂O and an oxygen gradient extending over about 2 nm from the top surface. Throughout this work, the thickness of Cu* refers to the total thickness of the Cu layer prior to oxidation. Relatively thick Cu* layers thus consist of an oxidized surface and a metallic Cu layer in contact with Co, as in Cu*(7)/Co, whereas Cu* layers thinner than 2 nm cannot prevent further oxygen penetration, leading to the partial

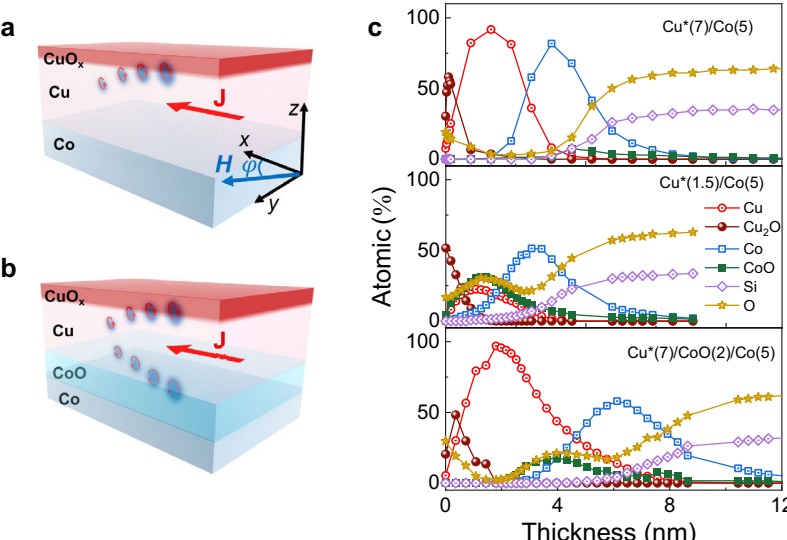

**Fig. 1 | Oxidation profiles in Cu and Co. a, b** illustrative schematics of the Cu*/Co and Cu*/CoO/Co heterostructures. **c** Relative concentration of elemental and oxide chemical species vs depth profiles determined by XPS combined with Ar⁺ etching of Cu*(7)/Co(5), Cu*(1.5)/Co(5), and Cu*(7)/CoO(2)/Co(5). The surface Cu₂O layer is due to natural oxidization in all three samples. The Si and O profiles at large thickness match the stoichiometry of the SiO₂ substrate.

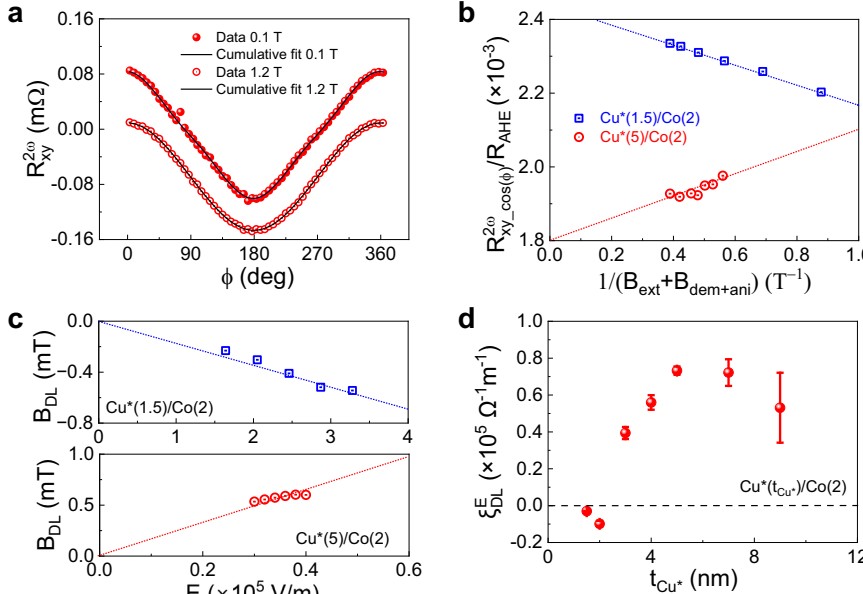

**Fig. 2 | Current-induced orbital torque in Cu*/Co. a** In-plane angular dependence of $R_{xy}^{2\omega}$ for Cu*(5)/Co(2) at external fields of 0.1 and 1.2 T. The solid lines are fits to Eq. (1). **b** The cos φ component of $R_{xy}^{2\omega}$ (denoted as $R_{xy\_\cos\varphi}^{2\omega}$) plotted against $1/(B_{ext} + B_{dem+ani})$. The applied electric field is $4 \times 10^4$ V/m (corresponding to a current of 10 mA) in Cu*(5)/Co(2) and $3.3 \times 10^5$ V/m in Cu*(1.5)/Co(2) (corresponding to a current of 4 mA). **c** Electric field dependence of $B_{DL}$ in Cu*(5)/Co(2) and Cu*(1.5)/Co(2). The positive or negative slope indicates a positive or negative torque, respectively. **d** Orbital torque efficiency as a function of Cu* thickness in Cu*($t_{Cu*}$)/Co(2). All data were measured at room temperature.

oxidation of Co. The oxidation of Co through Cu* as well as the deposition of the CoO spacer layer lead to an inverted oxygen gradient in the Cu* layer, as seen in the XPS profiles of Cu*(1.5)/Co(5) and Cu*(7)/CoO(2)/Co(5) in Fig. 1c. The formation of stoichiometric, anti-ferromagnetic CoO was confirmed by XPS and exchange bias measurements, as detailed in the Supplementary Fig. S2. CoO was found to be electrically insulating across the entire thickness range investigated in this study (Supplementary Fig. S3).

We used the harmonic Hall voltage method to measure the orbital torques in the Co layer generated by current flow in Cu*[62]. The samples were patterned into 20-μm-long, 10-μm-wide Hall bars using optical lithography and lift-off. We applied a sinusoidal voltage at a frequency $\omega/2\pi = 17$ Hz to induce an alternate current (AC) in the Hall bar and measured the first ($R_{xy}^{\omega}$) and second ($R_{xy}^{2\omega}$) harmonic components of the transverse resistance while rotating the sample in an external magnetic field. The torque-induced oscillations of the magnetization cause a second harmonic component in the Hall resistance due to the anomalous Hall and planar Hall effects, which is given by[62]

$$R_{xy}^{2\omega}(\varphi) = \left(\frac{1}{2}R_{AHE}\frac{B_{DL}}{B_{ext}+B_{dem+ani}} + R_{\nabla T}\right)\cos\varphi + R_{PHE}(2\cos^3\varphi - \cos\varphi)\frac{B_{FL}+B_{Oe}}{B_{ext}}$$

(1)

Here, φ is the angle of the in-plane external field $B_{ext}$ relative to the current direction, $R_{AHE}$ and $R_{PHE}$ are the anomalous Hall and planar Hall resistance coefficients, respectively, and $B_{DL}$, $B_{FL}$, and $B_{Oe}$ the current-induced effective fields resulting from the damping-like torque, field-like torque, and Oersted field, and $B_{dem+ani}$ the effective demagnetization and anisotropy field. Figure 2a shows $R_{xy}^{2\omega}$ of Cu*(5)/Co(2) measured in external fields of 0.1 and 1.2 T at room temperature for an applied AC current with a zero-to-peak amplitude of $I = 10$ mA. By fitting a series of these curves obtained for different values of $B_{ext}$ using Eq. 1, we obtain the coefficient of the first term, $R_{xy\_\cos(\varphi)}^{2\omega}$, which contains the damping-like torque. A linear fit of $R_{xy\_\cos(\varphi)}^{2\omega}$ vs $1/(B_{ext}+B_{dem+ani})$ further allows us to separate the thermal contribution to the second-harmonic resistance due to the spin Seebeck and Nernst effect, $R_{\nabla T}$, and derive $B_{DL}$ using the

values of $R_{AHE}$ and $B_{dem+ani}$ obtained from the measurement of $R_{xy}^{\omega}$ (see Supplementary Fig. S4).

Figure 2b compares $R_{xy\_\cos(\varphi)}^{2\omega}$ of Cu*(5)/Co(2) and Cu*(1.5)/Co(2), revealing a change of slope between the two samples. Repeating these measurements for different values of the applied current and electric field $E = IR_{xx}^{\omega}/L$, where $L$ is the length of the Hall bar and $R_{xx}^{\omega}$ its longitudinal resistance, we observe that $B_{DL}$ scales proportionally with $E$ in both samples, as shown in Fig. 2c. Notably, while $B_{DL}$ in Cu*(5)/Co(2) is positive, consistent with previous studies on Cu*/FM bilayers[3–10], $B_{DL}$ in Cu*(1.5)/Co(2) is negative. This change in sign is clearly associated with the differing oxidation level of the two samples, which is controlled by the thickness of the Cu* layer. To confirm this point, we measured $B_{DL}$ across samples with Cu* thickness ranging from 2 to 9 nm and calculated the torque efficiency per unit of applied electric field[3], $\xi_{DL}^E = \frac{2e}{\hbar}M_s t_{Co}B_{DL}/E$, where $e$ is the electronic charge, $\hbar$ the reduced Planck's constant, and $M_s t_{Co}$ the areal saturation magnetization of the Co layer (see Supplementary Fig. S5). The results, reported in Fig. 2d, show that $\xi_{DL}^E$ reaches a minimum of $-(9.9 \pm 1.7) \times 10^3$ $\Omega^{-1}$m$^{-1}$ in Cu*(2)/Co(2) and a maximum of $(7.3 \pm 0.3) \times 10^4$ $\Omega^{-1}$m$^{-1}$ in Cu*(5)/Co(2). Beyond $t_{Cu*} = 6$ nm, $\xi_{DL}^E$ decreases as more current is shunted through metallic Cu, which has negligible orbital and spin Hall conductivities.

In thicker Cu* samples, the positive $\xi_{DL}^E$ aligns with the expected sign of the electric-field induced orbital polarization of Cu* and the positive orbital-to-spin conversion taking place in the FM Co layer, where SOC favors parallel alignment of the spin and orbital magnetic moments. Conversely, the negative $\xi_{DL}^E$ observed in thinner Cu* samples is unexpected, indicating a reversal of either the orbital polarization or the orbital-to-spin conversion. The change of sign of $\xi_{DL}^E$ at a Cu* thickness between 2 and 3 nm coincides with the transition from oxidized to metallic Cu observed by XPS, suggesting that the partial oxidation of Co plays a role in altering the orbital damping-like torque.

## Insertion of CoO

To investigate the origin of the inversion of $\xi_{DL}$, we deliberately oxidized the top surface of the Co layer by exposure to oxygen in vacuum

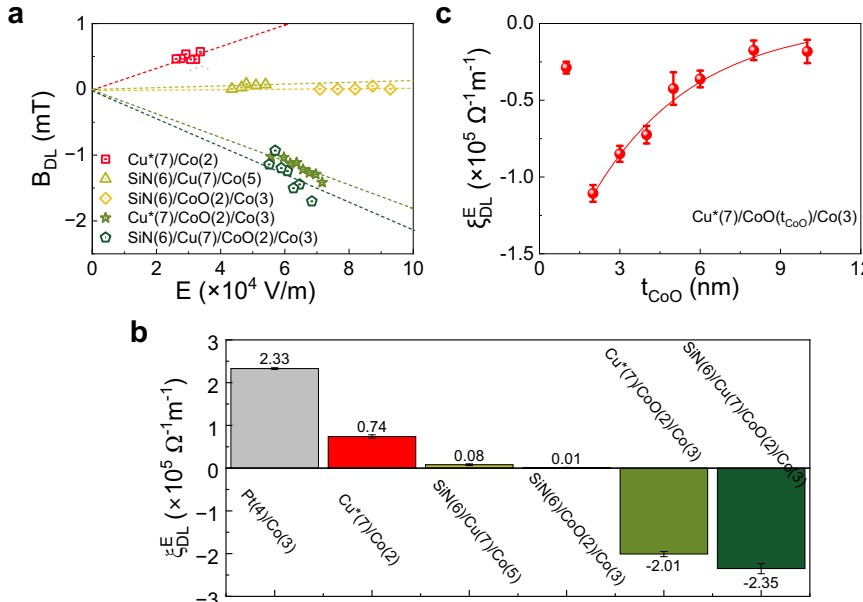

**Fig. 3 | Current-induced orbital torque in Cu*/CoO/Co and control samples.**
**a** Electric field dependence of $B_{DL}$ in different samples. The damping-like torque is negligible in SiN(6)/Cu(7)/Co(5) and SiN(6)/CoO(2)/Co(3), positive in Cu*(7)/Co(2), and negative in Cu*(7)/CoO(2)/Co(3) and SiN(6)/Cu(7)/CoO(2)/Co(3). **b** Comparison of the torque efficiency in the different samples and Pt/Co. In this series of samples, CoO was formed by post-deposition oxidation of Co in a controlled $O_2$ atmosphere. **c** Dependence of the orbital torque efficiency on CoO thickness in Cu*(7)/CoO($t_{CoO}$)/Co(3). In this series of samples, CoO was deposited from a stoichiometric CoO target. All data were measured at room temperature.

prior to the deposition of Cu, which resulted in a 2-nm-thick CoO layer. Figure 3a shows $B_{DL}$ as a function of $E$ measured in Cu*(7)/CoO(2)/Co(3) and Cu*(7)/Co(2), along with SiN(6)/Cu(7)/CoO(2)/Co(3) and the control samples SiN(6)/CoO(2)/Co(3), SiN(6)/Cu(7)/Co(5). The results indicate that $B_{DL}$ reverses sign upon the introduction of the CoO layer. Moreover, $B_{DL}$ is negligible in SiN(6)/CoO(2)/Co(3) and SiN(6)/Cu(7)/Co(5), ruling out significant contributions to the damping-like torque from the Rashba effect at the CoO/Co interface or from self-torque in the Co layer. Additionally, in SiN(6)/Cu(7)/CoO(2)/Co(3), which is protected from natural oxidation, $B_{DL}$ remains negative and has a similar magnitude to that in Cu*(7)/CoO(2)/Co(3), indicating that the inverted oxygen gradient at the Cu/CoO interface is the cause for the negative torque. Finally, as shown in Fig. 3b, the insertion of CoO increases the absolute value of $\xi_{DL}^E$ by up to a factor of three, resulting in a torque efficiency comparable to that of Pt/Co (see Supplementary Fig. S6).

Next, to better control the thickness of the CoO spacer and investigate the orbital transmission, we directly sputtered CoO from a stoichiometric target in a pure Ar atmosphere. Figure 3c shows $\xi_{DL}^E$ in the sample series Cu*(7)/CoO($t_{CoO}$)/Co(3), with $t_{CoO}$ varying from 1 to 10 nm. We observe an initial increase of $\left| \xi_{DL}^E \right|$ from 1 to 2 nm, followed by a monotonic decrease with increasing $t_{CoO}$. This behavior is expected upon the formation of a continuous insulating oxide layer. An exponential fit, consistent with a simple diffusive model of angular momentum propagation, gives $\xi_{DL}^E \sim e^{-t_{CoO}/\lambda}$, with decay length $\lambda = 3.8 \pm 0.9$ nm. This value agrees, within error, with the magnonic spin diffusion length of CoO, as measured in CoFeB/CoO/Pt trilayers via THz emission[63]. The decay length is significantly larger than the roughness of our layers (~0.4 nm) and the tunneling attenuation length of 0.1 nm reported for CoO thin films[64], suggesting that electron tunneling does not contribute in a significant way to orbital transport. We also find that the orbital torque in the samples with the sputtered CoO layers is lower than that observed when CoO was formed by surface oxidation, which we attribute to differences in the interfacial properties and oxygen concentration gradient in Co.

## Temperature dependence of the orbital torque

We studied the effect of magnetic ordering on the orbital torque to gain additional insight into the orbital-to-spin conversion mechanism mediated by CoO. In thin AFM films, magnetic ordering is a strong function of temperature and thickness. The Néel temperature of CoO reduces from $T_N = 293$ K in the bulk to below 200 K in 2-nm-thick CoO[65–67]. This reduction is also observed in our CoO films through the decrease of the blocking temperature $T_B$ estimated from the onset of exchange bias (see Supplementary Figs. S7 and S9), which occurs below but close to $T_N$ [65–67]. Figure 4a, b present the magnetic hysteresis loops of Cu*(7)/Co(2) and Cu*(7)/CoO(2)/Co(3) unpatterned films measured at 100 K after field cooling in a 0.3 T in-plane magnetic field. The magnetization of Cu*(7)/Co(2) is symmetric about zero field from room temperature down to 20 K, as seen in Fig. 4c, indicating the lack of exchange bias in this system, consistent with the absence of CoO evidenced by XPS in Fig. 1c. The horizontal shift ($B_{EB}$) and the increase of coercivity ($B_C$) of the magnetization loop of Cu*(7)/CoO(2)/Co(3) shown in Fig. 4d, on the other hand, confirm the AFM character of CoO. The negative $B_{EB}$ and its nearly-linear increase with decreasing temperature below $T_B \approx 200$ K (dashed line in Fig. 4d) are typical manifestations of exchange bias in CoO/Co[68,69]. Anisotropic magnetoresistance measurements performed on the patterned Hall bar samples confirm a similar temperature dependence of $B_{EB}$ as reported for the full films, and additionally show that $B_{EB}$ is strongly reduced by cycling the magnetic field after field cooling, consistent with the exchange bias training effects reported for CoO/Co[69].

Figure 4e, f show $\xi_{DL}^E$ as a function of temperature in the two samples. The torque efficiency of Cu*(7)/Co(2) and Cu*(7)/CoO(2)/Co(3) are both enhanced at low temperature. The 3-fold increase of $\xi_{DL}^E$ in Cu*(7)/Co(2) observed between 300 and 20 K is similar to that already reported for this system[10]. Such an increment of $\xi_{DL}^E$ has been attributed to the increase of the orbital Rashba-Edelstein effect resulting from the suppression of phonon-mediated orbital hybridization at low temperatures[70]. The behavior of $\xi_{DL}^E$ in Cu*(7)/CoO(2)/Co(3), however, is more complex. From room temperature to 200 K,

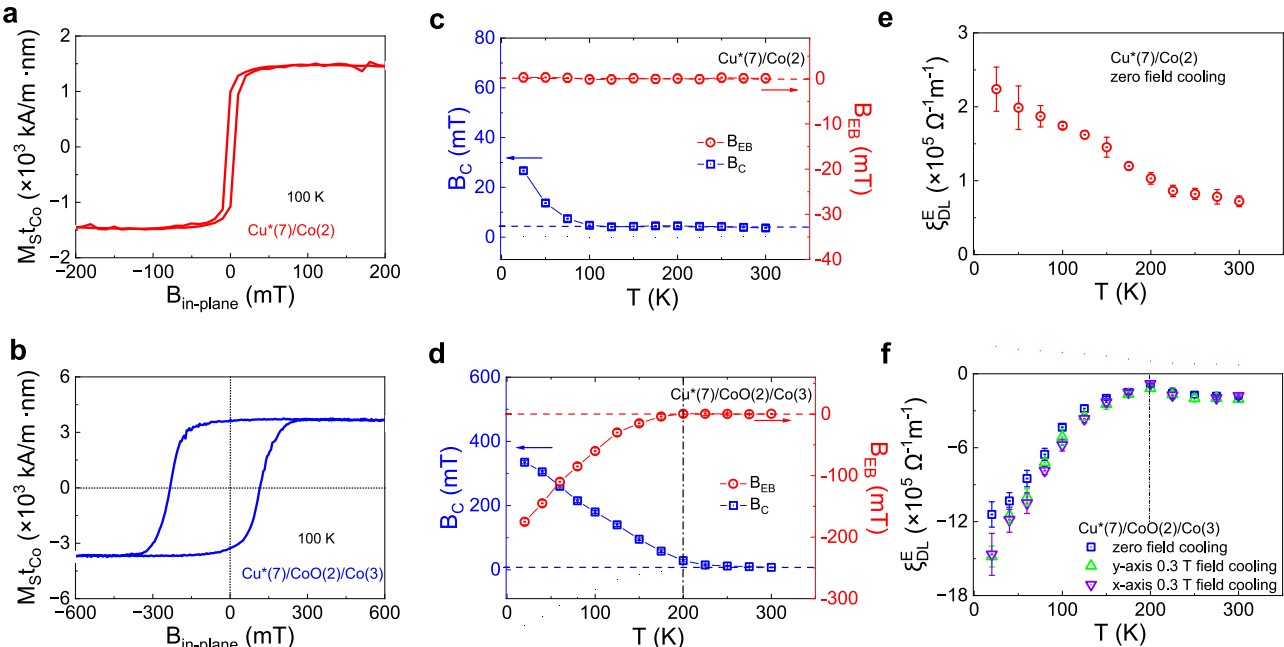

**Fig. 4 | Temperature dependence of the orbital torque and exchange bias.**
**a**, **b** Magnetic hysteresis loops of Cu*(7)/Co(2) and Cu*(7)/CoO(2)/Co(3), respectively, measured at 100 K after cooling in an in-plane magnetic field of 0.3 T. The presence of an exchange bias effect in Cu*(7)/CoO(2)/Co(3) confirms the AFM character of CoO. **c**, **d** Temperature dependence of the coercive field and exchange bias of Cu*(7)/Co(2) and Cu*(7)/CoO(2)/Co(3). The blocking temperature of Cu*(7)/CoO(2)/Co(3) is determined to be 200 K from the onset of exchange bias. **e**, **f** Orbital torque efficiency as a function of temperature in Cu*(7)/Co(2) and Cu*(7)/CoO(2)/Co(3). The different curves in (**f**) refer to zero-field cooling and cooling with field along the *x* or *y* axis (see Fig. 1a).

$\left|\xi_{DL}^{E}\right|$ decreases by about 50%. From 200 to 20 K, it increases by a factor 16 upon cooling in a field and by a factor 12 upon cooling in zero field. The maximum orbital torque efficiency at low temperature is $\xi_{DL}^{E} = -(1.48 \pm 0.09) \times 10^{6}\ \Omega^{-1}\mathrm{m}^{-1}$, substantially larger in magnitude compared to $\xi_{DL}^{E} = (2.2 \pm 0.3) \times 10^{5}\ \Omega^{-1}\mathrm{m}^{-1}$ measured in Cu*(7)/Co(2) without CoO spacer.

These observations reveal a striking increase in the orbital-to-spin transduction properties of CoO upon AFM ordering. The 30% enhancement of $\xi_{DL}^{E}$ in the field-cooled measurement relative to zero-field cooling is attributed to the increase of magnetic order at the CoO/Co interface. The alignment of orbital polarization relative to the direction of the in-plane field during cooling, on the other hand, does not impact $\xi_{DL}^{E}$ in a significant way, as shown in Fig. 4f.

## Discussion

We have demonstrated that the insertion of CoO between a non-magnetic Cu layer and a FM Co layer leads to both a sign reversal and a many-fold increase of the magnitude of the orbital torque transmitted to Co relative to a Cu*/Co bilayer. While the transmission of spin currents across iAFMs is well established[36–47], our findings reveal that CoO is an extremely efficient generator and transducer of orbital momentum. This process involves the generation of nonequilibrium orbital momenta at the Cu/CoO interface and the simultaneous transmission and conversion of angular momentum across CoO.

First, we discuss the sign reversal of the orbital torque observed upon oxidation of Co in both thin Cu*/Co and Cu*/CoO/Co samples. In Cu*, the current-induced orbital polarization arises from the asymmetric hybridization between O $p$-states and Cu $d$-states, driven by an oxygen gradient[30]. Reversing the oxygen gradient in Cu while keeping the current direction unchanged leads to a reversal of the orbital polarization. The change of sign of $\xi_{DL}^{E}$ observed in Cu*(1.5)/Co, Cu*/CoO/Co, and SiN/Cu/CoO/Co compared to Cu*(7)/Co, as reported in Fig. 2c, d and Fig. 3, aligns with the reversal of the oxygen gradient near the Cu/CoO interface

measured by XPS (Fig. 1). Similarly, we report a negative torque efficiency also in Cu*/NiO/Co (Fig. 5a), highlighting the importance of the inverted Cu/oxide interface in the generation of negative torque.

Next, we address the large increase in magnitude of the orbital torque in Cu*/CoO/Co and SiN/Cu/CoO/Co compared to Cu*/Co. The negligible torque detected in the control samples SiN/Cu/Co and SiN/CoO/Co confirms that both Cu and oxygen are required to generate a torque and rules out self-torques and the Rashba-Edelstein effect at the CoO/Co interface as possible explanations for it. Moreover, the analysis of the XPS spectra (Fig. 1c and Supplementary Fig. S2), reveals that oxygen preferentially binds to Co rather than Cu near the CoO interface, as expected from the larger enthalpy of formation of CoO relative to Cu$_2$O and CuO[71]. At room temperature, $\left|\xi_{DL}^{E}\right|$ in Cu*/CoO/Co and SiN/Cu/CoO/Co reaches values up to $2.35 \times 10^{5}\ \Omega^{-1}\mathrm{m}^{-1}$, about three times larger than in Cu*/Co (see Fig. 3b). This enhancement suggests that the diffuse interface between metallic Cu and CoO is a primary source of angular momentum in both systems. We attribute this behavior to an enhanced orbital Rashba-Edelstein effect, which is more prominent at the Cu/CoO interface compared to the Cu/Cu$_2$O interface typically found in Cu*[4,7].

The observation of a large torque further underscores the presence of effective mechanisms for transmitting and converting orbital momentum from the Cu/CoO interface to the Co layer. To investigate these mechanisms, we examine the temperature dependence of $\xi_{DL}^{E}$, as shown in Fig. 4f. The initial decrease of $\left|\xi_{DL}^{E}\right|$ from room temperature down to $T_B \approx 200$ K indicates a reduction in the transmission of angular momentum, consistently with a process mediated by the diffusion of thermal magnons. This behavior resembles the temperature-dependent decline in torque efficiency previously observed in the spin Hall systems FM/NiO/Pt (FM = CoFeB, Co)[48,51,55]. However, below $T_B$, an additional transmission channel opens up as the CoO layer orders antiferromagnetically, as evidenced by the onset of exchange bias and the concomitant enhancement of $\left|\xi_{DL}^{E}\right|$. This pronounced increase of $\left|\xi_{DL}^{E}\right|$ with the setting of antiferromagnetic order is corroborated by temperature-dependent measurements in a second batch of Cu*(7)/CoO($t_{CoO}$)/Co(3) trilayers with varying thickness of CoO, where the

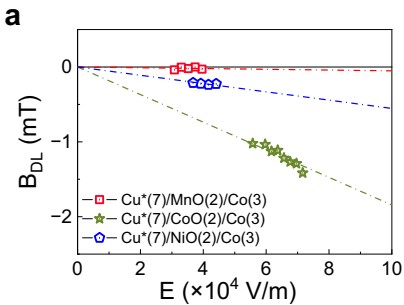

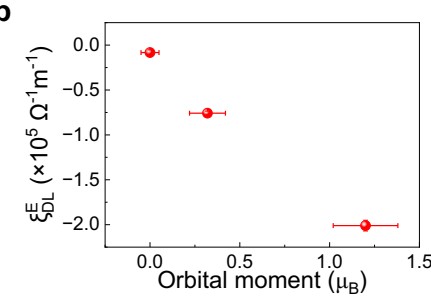

**Fig. 5 | Relation between orbital torque efficiency and orbital magnetic moment of CoO, NiO, and MnO. a** Electric field dependence of $B_{DL}$ in Cu*(7)/MnO(2)/Co(3), Cu*(7)/NiO(2)/Co(3) and Cu*(7)/CoO(2)/Co(3). **b** Orbital torque efficiency (room temperature values) as a function of the orbital magnetic moment of CoO, NiO, and MnO as determined by X-ray and neutron scattering measurements[82–85].

orbital torque consistently enhances below $T_B$ (see Supplementary Figs. S9 and S10). We also find that $\left|\xi^E_{DL}\right|$ decreases monotonically with increasing CoO thickness both above and below $T_B$, apart from samples with the thinnest CoO (1 nm), which present a significantly smaller torque efficiency likely due to a defective or discontinuous CoO layer (see Supplementary Figs. S9 and S10).

Previous studies have proposed three primary mechanisms through which an iAFM can transport angular momentum: (i) incoherent thermal magnons[36–40,45–51], (ii) coherent magnons[41–44,52–54,72], and (iii) a spin-flop effect mediated by exchange bias[55,73]. These processes are not mutually exclusive, as demonstrated by the temperature-dependent sign inversion of spin torque and spin Hall magnetoresistance in FM/NiO/Pt trilayers[46,48,49,55]. Incoherent thermal magnons, however, cannot account for the strong increase of $\left|\xi^E_{DL}\right|$ observed below $T_B$ in Cu/CoO/Co. Additionally, $\xi^E_{DL}$ remains unchanged if the exchange bias in CoO is set parallel (along $y$) or perpendicular (along $x$) to the injected orbital polarization (see Fig. 4f), while the orbital Rashba-Edelstein magnetoresistance, which is analog to the spin Hall magnetoresistance in FM/NiO/Pt, retains its sign above and below $T_B$ (see Fig. S11). These observations indicate that the spin flop mechanism cannot explain the large orbital torque mediated by CoO.

Most critically, the orbital nature of the torque implies that none of the aforementioned spin-based mechanisms—whether involving magnons or exchange bias—can account alone for the orbital-to-spin conversion mediated by CoO. Efficient orbital-to-spin conversion in NM/FM bilayers typically requires a relatively thick FM with spin-orbit split electronic states near the Fermi level, such as Ni. Alternatively, heavy metal spacers[3,12,13,16] provide the large SOC required for the conversion. In Cu*/CoO/Co, the insulating character of CoO and the giant value of $\left|\xi^E_{DL}\right| \approx 1.5 \times 10^6\ \Omega^{-1}m^{-1}$ reached at low temperature point towards a different and extremely efficient orbital-to-spin conversion mechanism. Here, we postulate that such a mechanism is related to the unquenched orbital magnetic moment and entangled spin-orbital character of the low-energy Co states in CoO.

In the weak-to-intermediate crystal field limit relevant for CoO, the $^4F$ free-ion ground state of Co$^{2+}$ ($S = 3/2$, $L = 3$) in octahedral symmetry branches in a 12-fold degenerate $^4T_{1g}$ orbital triplet. Within this triplet, the spin remains $S = 3/2$ and the orbital moment is described by $\boldsymbol{L} = -(3/2)\boldsymbol{l}$, where $\boldsymbol{l}$ is an effective orbital momentum operator with $l = 1$. SOC further separates the orbital triplet states in three effective total angular momentum $\boldsymbol{j}_{eff} = \boldsymbol{l} + \boldsymbol{S}$ manifolds corresponding to $j_{eff} = \frac{1}{2}, \frac{3}{2}$ and $\frac{5}{2}$ [56,74]. These many-electron states can be expressed as combinations of single-particle states, with dominant contributions from the $\left(t^\uparrow_{2g}\right)^3\left(e^\uparrow_g\right)^2\left(t^\downarrow_{2g}\right)^2$ state[75,76]. Upon cooling below the Néel temperature, CoO undergoes a tetragonal lattice distortion concurrently with the onset of type-II antiferromagnetic order. This magnetic transition is mediated by superexchange interactions involving both nearest and next-nearest neighbor Co ions. The distortion of the crystal field and time-reversal symmetry breaking by magnetic exchange lead to substantial mixing of the spin-orbit split levels, removing the residual degeneracy of the $j_{eff}$ manifolds[57–59,77–79]. Thus, below the Néel temperature, the admixture of spin-orbit split crystal field levels in CoO results in a complex low-energy magnetic excitation spectrum that is best described by a spin-orbit exciton model[59,79]. The term exciton here refers to an intra-atomic transition between the spin-orbit coupled electronic states of CoO, akin to those found in iridate compounds such as Sr$_2$IrO$_4$[80]. Excitons change not just the spin or orbital state, but the combined total angular momentum of an ion. Despite their localized character, the superexchange interaction between neighboring ions allows a spin-orbit exciton to propagate and interact with the spin degree of freedom, effectively hybridizing with coherent magnon excitations[57,59,79,81]. Evidence for coupled spin-orbit excitons and magnon modes in CoO has been reported by inelastic neutron scattering[56,58–79,81], Raman[77], and optical spectroscopy[78].

Based on these findings, we believe that the most plausible interpretation of our data involves the localized excitation of spin-orbit excitons by electron scattering at the Cu/CoO interface, their propagation via magnetic exchange through the thin CoO layer, and final relaxation via coupling to the spin degree of freedom of metallic Co, leading to a spin torque. The spin-orbit exciton process conserves total angular momentum[57,78,81] and thereby allows for orbital momentum to be converted to $j_{eff}$ and transported across CoO. The efficiency of this process increases at low temperature as the establishment of long-range antiferromagnetic order increases both the efficiency of the orbital-to-spin conversion process—by inducing the admixture of different spin-orbit split crystal field manifolds—and the propagation of spin-orbital excited states via collective magnetic dynamics, consistently with our measurements. Above the Néel temperature, the conversion is less efficient and the transport is mediated by incoherent magnetic fluctuations, leading to a moderate torque efficiency that increases with temperature.

Although a detailed theoretical modeling of these processes goes beyond the scope of our work, indirect support for the spin-orbit exciton scenario can be gained by the comparison of the orbital torque efficiency mediated by X = CoO, NiO, and MnO in Cu*(7)/X(2)/Co(3) trilayers. These isostructural monoxides share the same type II antiferromagnetic spin lattice. However, unlike orbitally-degenerate Co$^{2+}$, the ground state of Ni$^{2+}$ in octahedral symmetry is an orbital singlet with a small admixture of higher lying states induced by SOC, and the ground state of Mn$^{2+}$ is a pure orbital singlet. Room temperature torque measurements, reported in Fig. 5, show that $\left|\xi^E_{DL}\right|$ is significantly reduced in Cu*/NiO/Co relative to Cu*/CoO/Co, and vanishingly small in Cu*/MnO/Co. Furthermore, $\left|\xi^E_{DL}\right|$ remains constant in Cu*/MnO/Co down to a temperature of 10 K, i.e., well below $T_B$ of MnO (Supplementary Fig. S12). Most interestingly, the orbital torque efficiency

scales with the orbital magnetic moment localized on the transition metal ions, ranging from $m_L \approx 1\,\mu_B$ in CoO[82–84], to 0.3 $\mu_B$ in NiO[83,85], and $\approx 0$ in MnO[83]. This comparison shows that iAFMs with coupled spin and orbital degrees of freedom are most effective in converting and transporting orbital momentum, confirming that the two processes are related. Phenomenologically, these processes can be described by a temperature-dependent spin-orbital mixing conductance[13,32].

In perspective, our results show that iAFM can be used not only to transport spin currents, as shown using both local and nonlocal transmission geometries[37–42], but also to generate, transport, and convert orbital currents. The high conversion efficiency of CoO further demonstrates that transition-metal oxides with unquenched orbital moments are very promising materials for boosting the efficiency of orbital torques without requiring thick FM layers. Recent studies have demonstrated that orbital torques can switch the magnetization of FM layers in devices[12,86–89]. Since the switching current is inversely proportional to the torque efficiency, and exchange bias in AFM can be used to promote field-free switching[90,91], our findings suggest that iAFMs with an orbitally degenerate ground state and entangled spin-orbital excitations can be used to significantly lower the critical switching current of magnets with perpendicular magnetic anisotropy.

## Methods
### Sample fabrication
The thin film heterostructures were fabricated using magnetron sputtering at a base pressure of $8 \times 10^{-8}$ mbar on Si/SiO$_2$ substrates at room temperature, with an argon pressure of $2.3 \times 10^{-3}$ mbar. The deposition rates for Co, Cu, and CoO are 40 s/nm, 15 s/nm, and 39 s/nm, respectively. These rates were calibrated by measuring the step height with an Atomic Force Microscope for films deposited over a specific sputtering duration. The samples with Cu were naturally oxidized in the air for 2 days after fabrication; the 3 nm CuO$_x$ on top prevents the further oxidization of the stacks. The CoO/Co bilayers were fabricated either by introducing 100 sccm Ar and 4 sccm O$_2$ for 10 min after the deposition of Co or by depositing CoO on top of the Co layer from a stoichiometric CoO target. The standard photolithography and lift-off are carried out to fabricate the Hall bars for subsequent electrical measurement.

### XPS measurements and Ar$^+$ sputtering depth profile
Small-area XPS was performed to identify the chemical state of Cu, Co, and O in the Cu*(7)/Co(5), Cu*(1.5)/Co(5), and Cu*(7)/CoO(2)/Co(5) samples[92,93]. The in-depth distribution of the elements was further examined through Ar$^+$ ion sputtering in conjunction with XP-spectra acquisition. A monochromatic AlKα source with an energy of 1486.6 eV was operated at the nominal spot size of 400 nm. The calibration and linearity of the binding energy scale were checked before the sample analysis, following ISO 15472: 2010. The analyzer was used in the fixed-analyzer transmission mode. Pass energies used for survey and detailed scans were 200 eV and 100 eV, respectively, in standard lens mode. With the selected scan parameters, the energy resolution, as measured by the FWHM of the Ag 3d$_{5/2}$ peak acquired with a 100 eV pass energy on a sputter-cleaned Ag sample during the calibration, was 0.97 eV. To acquire the compositional depth profile, the samples were Ar-sputtered at a 45° incidence angle using a sputtering rate of 1.1 nm per 60 s calibrated on a SiO$_2$/Si wafer. The thickness of each layer is calculated as the difference between the two inflection points of the dept profile. For example, in the case of Cu*(7)/CoO(2)/Co(5) (bottom panel of Fig. 1c), the thickness of the Co layer is estimated to be 8.5–4 nm = 4.5 nm, considering the Co profile at the interfaces with Co/Si and CoO/Co. The XPS data were processed using the CasaXPS software[94]. A Shirley-Sherwood background was subtracted before fitting the spectra with the parameters reported in Table S1 in the Supplementary Fig. S2. The survey and the high-resolution spectra are reported in the Supplementary Fig. S2.

### Magnetization measurements
Magnetization data were collected on blanket layers using a superconducting quantum interference device (SQUID) magnetometer (Quantum Design). The areal saturation magnetization $M_s t_{Co}$ of various samples was determined through magnetic hysteresis loop measurements. For the exchange bias measurement, the samples were cooled to 20 K while applying a magnetic field of 0.3 T from room temperature, and hysteresis loops were recorded at each temperature during the heating process. The exchange bias field was calculated by averaging the positive and negative coercive fields. The results presented in Fig. 4 are the first hysteresis loops without any training effect. The exchange bias field $B_{EB}$ can also be determined in the Hall bar samples by fitting the anisotropic magnetoresistance after field cooling in Hall bar devices (refer to Supplementary Fig. S13). Due to geometrical confinement and training effects, the values of $B_{EB}$ obtained in this way are reduced to less than 20% of $B_{EB}$ obtained from the SQUID measurements in full films before training.

### Orbital torque measurements
The orbital torques were measured on patterned samples using the harmonic Hall voltage method[62] as described in the main text. The measurements reported in Figs. 2 and 3 were performed at room temperature in magnetic fields of up to 1.6 T with ac currents ranging from $I = 14$ to 18 mA. The measurements reported in Fig. 4 were performed in a variable temperature setup in magnetic fields of up to 0.97 T. The samples were thermalized in a He atmosphere before starting the measurements. The sample resistance was measured at each temperature to calculate the applied electric field $E = IR_{xx}^{\omega}/L$ from the applied current. We note that in the presence of exchange bias Eq. (1) should be modified to include $B_{EB}$ in the denominator of the first term. When applying Eq. (1) to samples with a net $B_{EB}$ after field-cooling, we took care that such a correction remained negligible by considering measurements acquired for $B_{ext} \gg B_{EB}$, where $B_{ext} \geq 600$ mT and $B_{EB} \lesssim 30$ mT.

## Data availability
All data are available in the main text or the supplementary materials from ETH data collection https://doi.org/10.3929/ethz-b-000703564.

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

## Acknowledgements

This work is supported by the Swiss National Science Foundation (Grant No. 200020_200465). P.N. acknowledges the support of the ETH Zurich Postdoctoral Fellowship Programme (19-2 FEL-61).

## Author contributions

S.D. and P.G. designed and supervised this study. S.D. performed the electrical measurement. S.D. analyzed the data with the inputs from P.N., G.-K.K., D.N., and P.G. G.S. contributed to the device fabrication and magnetic characterization. A.R. and M.F. performed the X-ray photoelectron spectroscopy measurements and analysis. S.D. and P.G. wrote the paper. All authors discussed the results and commented on the paper.

## Competing interests

The authors declare no competing interests.
