## [Peer review File · Nature Communications]

Generation, Transmission, and Conversion of Orbital Torque by an Antiferromagnetic Insulator

Corresponding Author: Dr Shilei Ding

Version 0:

Reviewer comments:

Reviewer #1

(Remarks to the Author)

The authors study the generation and conversion of orbital currents, as well as the resultant orbital torque in CuOx/CoO/Co samples. They observed a sign change in orbital torque upon the insertion of CoO into CuOx/Co. Their results show that CoO is an effective orbital-to-spin transduction. The efficiency of orbital torque in CuOx/CoO/Co is comparable to that in Pt/Co and exhibits a pronounced increase at low temperatures, suggesting a potential connection with the magnetic order in CoO. The findings reported in this manuscript are both surprising and intriguing, as they introduce a novel concept of converting and transmitting orbital momentum by an insulating AFM, representing an advance in both physics and applications.

However, I cannot recommend the publication of this paper in its present form. The evidence presented in the manuscript is insufficient to support the conclusions. More importantly, the authors have not provided a reasonable model or interpretation of the conversion and transmission of orbital momentum by insulating CoO, which is the key novelty of this work. The following issues must be addressed clearly in the manuscript.

- (1) The conversion and transmission of orbital momentum are two completely different effects. One possible process could be: the orbital momentum is converted to spin momentum at the CuOx/CoO interface. Then the spin momentum penetrates through the CoO layer and exerts a torque on Co moment. The alternative process could be: the orbital momentum directly penetrates through the CoO layer and is converted to spin momentum in the Co layer. The authors should specify which process is prominent in their results. And what kind of phenomenon CoO is involved in.
- (2) As proposed by the authors in the manuscript, the orbital-to-spin conversion could be related to the orbital triplet ground state of the Co²⁺ ions in CoO. As a major novelty of this work, the authors should provide a detailed theoretical interpretation of this mechanism if it is true.
- (3) The authors also propose that orbital angular momentum can be transferred to the CoO magnons through electrons scattering. If the CoO magnons are indeed the carriers of angular momentum, the effect of orbital torque should be suppressed at low temperatures, as reported in several previous studies (Z. Qiu et al., Nature Commun. 7, 12670 (2016)). However, this conjecture contradicts the experimental result depicted in Figure 4f.
- (4) In order to verify the contribution of the CoO magnons at low temperature, the CoO thickness dependence of the orbital torque efficiency must be determined. The CoO thickness dependence is not a simple exponential decay if contributed by the coherent AFM magnons (Roman Khymyn et al., Phys. Rev. B 93, 224421 (2016)).
- (5) It should be emphasized that the transmission of orbital angular momentum in an AFM insulator is very challenging (Randy S. Fishman et al., Phys. Rev. Lett. 129, 167202 (2022)). In proposing the spin-orbital mixing conductance across the CoO layer, the authors must provide a clear interpretation of why the transmission of orbital angular momentum in an insulating CoO is possible.
- (6) In order to incorporate the spin-flop coupling and the exchange spring effect into the interpretation, it is necessary to measure the coupling at the CoO/Co interface. For instance, spin Hall magnetoresistance (Dazhi Hou et al., Phys. Rev. Lett. 118, 147202 (2017)).
- (7) The comparison of Cu*/CoO/Co, Cu*/Co and SiN/Cu/CoO/Co is not quite convincing. The interfaces are not adequately controlled in this comparison. To substantiate the assertion that the sign change in the orbital torque is attributable to an inverted oxygen gradient, more direct evidence is necessary.

Reviewer #2

(Remarks to the Author)

The generation, transmission, and conversion of orbital torque are very important for the electrical control of magnetization in nanoscale devices. In this paper, the authors demonstrate a large orbital torque and enhanced orbital-to-spin conversion by an antiferromagnetic insulating CoO layer. They found that inserting CoO in CuOx/CoO/Co trilayers reverses the sign of torque and triples its magnitude compared to the CuOx/Co, which is explained by the inverted oxygen gradient at the CuOx/CoO as compared with the Cu/Co interface. Moreover, at low temperatures, the onset of antiferromagnetic order and exchange bias induces a further many-fold increase of the torque. These results reveal that antiferromagnetic insulators like CoO can provide highly effective orbital-to-spin transduction to improve the performance of spintronic devices. The experimental results are very interesting, and the analysis on the experimental results is very reasonable. However, some issues should be clarified before its publication in NC.

1. In 102-104 lines : Elemental depth profiling by X-ray photoelectron spectroscopy (XPS) shows that natural oxidation of Cu leads to the formation of Cu₂O and an oxygen gradient extending over about 2 nm from the top surface, see Fig. 1c. In 385-386 lines: The samples with Cu were naturally oxidized in the air for two days after fabrication, the 3 nm CuOx on top prevents the further oxidization of the stacks. "An oxygen gradient extending over about 2 nm from the top surface" may be in conflict with "the 3 nm CuOx on top".
2. In 116-118 lines, Fig.1 c shows the relative concentration of elemental and oxide chemical species vs depth profiles determined by XPS combined with Ar⁺ etching of Cu^{*(7)}/Co(5), Cu^{*(1.5)}/Co(5), and Cu^{*(7)}/CoO(2)/Co(5). The peak positions of Cu and Co in Fig.1 c seem contradictory to the thickness of the Cu and Co layers. For example, in the bottom panel of Fig.1c, the Co peak position of Cu^{*(7)}/CoO(2)/Co(5) is located at about 6.5 nm. How thick is the detecting depth of the XPS ?
3. In 34-36 Lines, the authors mentioned that similar to spin torques, orbital torques can be used for magnetization switching, domain wall motion, and excitation of high-frequency dynamics in nano-oscillators. One wonders whether the enhanced orbital torques can switch the magnetization of Co layer in their samples. If not, please give a brief discussion on it.

Reviewer #3

(Remarks to the Author)

In this work, the authors showed a notable orbital torque in CuOx/CoO/Co trilayer, where the orbital current generated by CuOx passed through and converted to spin current in CoO layer, resulting in a sizable torque efficiency. This work is interesting, and may open a possibility that using insulating antiferromagnets as the spacers for orbital-to-spin conversion. I can recommend the publication if the concerns below are addressed:

1. The authors suggested that CoO layer is insulating. They should provide evidence that the thin CoO layer is indeed insulating, otherwise CoO layer might be just like an ordinary metallic orbital-to-spin converter.
2. The authors showed that field cooling along different directions does not influence the results. However, the field applied is too small, which might not be sufficient to influence the AFM state. I suggest the authors check the results using a much larger cooling field.
3. Since there is an exchange bias, it is possible to realize an efficient field free switching of perpendicular magnetization. It would be good to show this in experiment, or at least add some discussion to mention this possibility.

Version 1:

Reviewer comments:

Reviewer #1

(Remarks to the Author)

The authors made great efforts to address potential issues. Their additional experimental results, e.g., the dependence of the torque efficiency on the AFM materials (CoO, NiO, and MnO), are critical evidence that supports their conclusions. Such evidence can provide useful information to the community. The authors' response has addressed most of my concerns. Currently, my only concern is whether there is any direct evidence confirming that the torque exerted on the Co moment is produced by the orbital current through the CoO. Although the control experiments with CoO, NiO, and MnO are quite convincing, they are indirect. Direct evidence could be the dependence of torque efficiency on the thickness of the Co layer. However, the outcome of this control experiment is unpredictable. And I understand that the control experiment is time-consuming. Hence the indirect evidence might be sufficient to support their main claim. Given the experiment results they provided are indeed intriguing and could advance this topic, I suggest that the authors balance the timeliness and soundness of this work. And there are no more comments.

Reviewer #2

(Remarks to the Author)

The authors have carefully revised the manuscript and answered all questions. The quality of the paper is greatly enhanced. I would like to recommend its publication in NC.

Reviewer #3

(Remarks to the Author)

The authors have addressed my concerns, and I recommend the publication of this work.

Reviewer #1 (Remarks to the Author):

The authors study the generation and conversion of orbital currents, as well as the resultant orbital torque in CuOx/CoO/Co samples. They observed a sign change in orbital torque upon the insertion of CoO into CuOx/Co. Their results show that CoO is an effective orbital-to-spin transduction. The efficiency of orbital torque in CuOx/CoO/Co is comparable to that in Pt/Co and exhibits a pronounced increase at low temperatures, suggesting a potential connection with the magnetic order in CoO. The findings reported in this manuscript are both surprising and intriguing, as they introduce a novel concept of converting and transmitting orbital momentum by an insulating AFM, representing an advance in both physics and applications.

However, I cannot recommend the publication of this paper in its present form. The evidence presented in the manuscript is insufficient to support the conclusions. More importantly, the authors have not provided a reasonable model or interpretation of the conversion and transmission of orbital momentum by insulating CoO, which is the key novelty of this work. The following issues must be addressed clearly in the manuscript.

We thank the referee for reviewing our work and for recognizing both its conceptual and practical novelty. We greatly appreciate the critical comments, as well as the opportunity to further elaborate on the physical mechanisms underpinning the conversion and transmission of orbital angular momentum by antiferromagnetic CoO.

Comments (1) and (2) address theoretical aspects that are central to the interpretation of our results. In this respect, we should say upfront that developing a comprehensive theoretical model of orbital conversion and transmission in a strongly correlated antiferromagnetic oxide like CoO lies beyond the scope of our expertise, which is primarily experimental. While a collaboration with theorists could eventually lead to a suitable model, we believe that presenting our experimental findings—even in the absence of a definitive theoretical model—will stimulate interest and discussion within the broader theoretical community, allowing different models to emerge and be compared. Despite this premise, we believe that we can now provide a consistent qualitative framework to interpret our findings, which is corroborated by new measurements performed on NiO and MnO samples, in addition to CoO.

In order to address the criticism and comments by the referee, we have taken the following actions:

1. We have fabricated new samples and performed measurements on other antiferromagnetic oxides with varying degrees of orbital degeneracy (NiO, MnO) to reinforce the robustness of the experimental conclusions and provide arguments that support the identification of spin-orbital excitations in CoO as the main mechanism for orbital conversion and transport.
2. We have substantially revised and extended the discussion section in order to clarify the possible mechanisms responsible for the conversion and transmission

of orbital angular momentum in antiferromagnetic CoO, elaborating on the type of magnetic excitations in CoO compatible with our findings.

We hope that these efforts address the referee's concerns and improve the clarity and impact of our work. A point-by-point response to the referee's criticism can be found below.

(1) The conversion and transmission of orbital momentum are two completely different effects. One possible process could be: the orbital momentum is converted to spin momentum at the CuOx/CoO interface. Then the spin momentum penetrates through the CoO layer and exerts a torque on Co moment. The alternative process could be: the orbital momentum directly penetrates through the CoO layer and is converted to spin momentum in the Co layer. The authors should specify which process is prominent in their results. And what kind of phenomenon CoO is involved in.

We thank the referee for the question, which gives us the opportunity to further discuss the observed results. Because of the large orbital moment and non-negligible spin-orbit coupling of CoO, the spin and orbital degrees of freedom in this material cannot be separated. Therefore, the conversion and transport of orbital momentum can be considered to be part of the same process. Electrons with finite orbital polarization generated in CuOx that scatter at the CoO interface induce spin-orbit excitons rather than just spin flips or orbital flips. These spin-orbital excitations can then propagate through CoO via magnetic exchange and couple to the (mostly) spin degree of freedom of metallic Co, leading to a spin torque. In other words, we believe that the two processes outlined by the referee are entangled in CoO.

There are several works stretching over the last decades discussing the mixed spin-orbital character of collective magnetic excitations – such as spin-orbit excitons and magnons – in CoO, which has a quite complex low-energy spectrum (see Refs. 74-80 in the revised manuscript and references therein). We are not aware, however, of theoretical models describing the conversion of an orbital current into mixed spin-orbital excitations. Even the simpler conversion of an orbital current into a spin current in model metal systems has been theoretically modelled only in the bulk rather than at interfaces (see Ref. 2 in the revised manuscript).

Despite the lack of theoretical models to describe orbital-to-spin conversion and propagation in spin-orbit coupled systems, the point raised by the referee can also be addressed by experiments. The relevance of mixed spin-orbital excitations is supported by comparing the torques measured in CuOx/CoO/Co with new measurements performed in CuOx/AFM/Co samples, where AFM = NiO and MnO. These monoxides share the same rock-salt crystal structure and antiferromagnetic type II spin lattice, but have quite different orbital magnetization localized on the transition metal ions, ranging from $m_L \approx 1 \mu_B$ in CoO, to $0.3 \mu_B$ in NiO, and zero in MnO. The torque measurements, reported in Fig. R1 below, show that the orbital torque efficiency scales with m_L , being largest in CuOx(7)/CoO(2)/Co(3) and vanishing in

CuOx(7)/MnO(2)/Co(3). This comparison shows that antiferromagnets with coupled spin and orbital degrees of freedom are most effective in converting and transporting orbital momentum, suggesting that the two processes are related. We have added the new data and a revised discussion to the manuscript.

Figure R1. Relation between orbital torque efficiency and the orbital magnetic moment of the antiferromagnetic insulator. **a**, Electric field dependence of B_{DL} in Cu*(7)/MnO(2)/Co(3), Cu*(7)/NiO(2)/Co(3) and Cu*(7)/CoO(2)/Co(3). **b**, Dependence of orbital torque efficiency on the orbital magnetic moment, with the orbital moment values obtained from previous experimental studies (see manuscript for references).

(2) As proposed by the authors in the manuscript, the orbital-to-spin conversion could be related to the orbital triplet ground state of the Co²⁺ ions in CoO. As a major novelty of this work, the authors should provide a detailed theoretical interpretation of this mechanism if it is true.

As mentioned above, detailed theoretical models of orbital-to-spin conversion processes at interfaces and in strongly-correlated materials are currently unavailable. However, it is possible to outline how such a process could take place by considering the spectrum of electronic and magnetic excitations of Co ions in CoO.

CoO has a cubic NaCl crystal structure, where the Co²⁺ ions occupy sites with octahedral symmetry. In the weak-to-intermediate crystal field limit relevant for CoO, the ⁴F free-ion ground state of Co²⁺ ($S = 3/2$, $L = 3$) branches in a 12-fold degenerate ⁴T_{1g} orbital triplet. Within this triplet, the spin remains $S = 3/2$ and the orbital moment is described by $\mathbf{L} = -(3/2)\mathbf{l}$, where \mathbf{l} is an effective orbital momentum operator with $l = 1$. Spin-orbit coupling further separates the orbital triplet states in three effective total angular momentum $\mathbf{j}_{\text{eff}} = \mathbf{l} + \mathbf{S}$ manifolds corresponding to $j_{\text{eff}} = \frac{1}{2}, \frac{3}{2}$ and $\frac{5}{2}$ (Refs. 74 and 75 in the revised manuscript). These many-electron states can be expressed as combinations of single-particle states, with dominant contributions from the $(t_{2g}^{\uparrow})^3 (e_g^{\uparrow})^2 (t_{2g}^{\downarrow})^2$ state (Refs. 76 and 77 in the revised manuscript). Upon cooling below the Néel temperature, CoO undergoes a tetragonal lattice distortion concurrently with the onset of type-II antiferromagnetic order. This magnetic transition is mediated by superexchange interactions involving both nearest and next-nearest neighbor Co ions. The distortion of the crystal field and time-reversal symmetry

breaking by magnetic exchange lead to substantial mixing of the spin-orbit split levels, removing the residual degeneracy of the j_{eff} manifolds (Refs. 57-59 and 78-80 in the revised manuscript).

The strong entanglement of spin-orbit split crystal field levels in CoO results in a complex low-energy magnetic excitation spectrum that is best described by a spin-orbit exciton model (Refs. 59 and 80 in the revised manuscript). The term exciton here refers to an intra-atomic transition between the spin-orbit coupled electronic states of CoO, akin to those found in $4d$ or $5d$ transition-metal oxides with strong spin-orbit coupling, such as Sr_2IrO_4 (Ref. 81 in the revised manuscript). Excitons change not just the spin or orbital state, but the combined total angular momentum of an ion. Despite their localized character, the superexchange interaction between neighboring ions enables virtual hopping processes that allow a spin-orbit exciton to propagate and interact with the spin degree of freedom, effectively hybridizing with coherent magnon excitations (Refs. 57,59,80 and 82 in the revised manuscript). Evidence for coupled spin-orbit excitons and magnon modes in CoO has been reported by inelastic neutron scattering (Refs. 80 and 82 in the revised manuscript), Raman (Ref. 78 in the revised manuscript) and optical spectroscopy (Ref. 79 in the revised manuscript). These studies identified several weakly-dispersing magnetic-dipole transitions between spin-orbit split levels of CoO in the 15-40 meV energy range, whose intensity vanishes at T_N .

Based on these findings, we believe that the most plausible interpretation of our data – in particular of the large torque efficiency observed below the Néel temperature - involves the localized excitation of spin-orbit excitons by electron scattering at the CuOx/CoO interface, their propagation via magnetic exchange interactions through the thin CoO layer, and final relaxation via spin transfer torque in metallic Co. The spin-orbit exciton process conserves total angular momentum (Refs. 57,59 and 82 in the revised manuscript) and thereby allows for orbital momentum to be converted to j_{eff} and transported across CoO. The efficiency of this process increases at low temperature as the establishment of long-range antiferromagnetic order mixes the different j_{eff} levels and facilitates the propagation of single-ion excited states via collective magnetic dynamics. Above the Néel temperature, on the other hand, incoherent magnons (hybridized with spin-orbital excitations) can also facilitate angular momentum transfer across CoO, leading to a torque efficiency that increases with temperature, consistently with the reported experimental trend.

Although we cannot provide a detailed theoretical modelling of these processes, our measurements of the torque efficiency in isostructural systems with varying degrees of spin-orbit coupling support this explanation.

(3) The authors also propose that orbital angular momentum can be transferred to the CoO magnons through electrons scattering. If the CoO magnons are indeed the carriers of angular momentum, the effect of orbital torque should be suppressed at low temperatures, as reported in several previous studies (Z. Qiu et al., Nature Commun.

7, 12670 (2016)). However, this conjecture contradicts the experimental result depicted in Figure 4f.

We thank the referee for this question, which prompted us to restructure the discussion of our results. Indeed, measurements of spin transmission mediated by insulating antiferromagnetic oxide layers such as NiO and CoO show that the spin pumping voltage (Qiu & Saitoh, NC 2016; Li & Qiu, NC 10 5265 2019) and spin Seebeck effect (Ref. 42 in the revised manuscript) peak at $T \approx T_N$ and decay at $T < T_N$, as expected for processes that rely on thermally populated magnons. For spin-orbit torque mediated by antiferromagnets, however, there are multiple mechanisms that can lead to spin transport across antiferromagnetic insulators, ranging from incoherent thermal magnons at high temperature (Refs. 36-40 and 45-51 in the revised manuscript) to coherent magnetic excitations (Refs. 41-44, 52-54 in the revised manuscript) and interfacial spin-flop coupling (Refs. 55 in the revised manuscript) at low temperature. NiO is an exemplary case in which a nonmonotonic temperature dependence of the spin torque efficiency has been reported due to the coexistence of these different processes (Refs. 55 in the revised manuscript).

In CuOx/CoO/Co, upon lowering the temperature, the torque reaches a minimum at the Néel temperature and then increases again at lower temperatures (Fig. 4f, see zoom-in below). This behavior is consistent with a torque being mediated by localized spin-orbit excitons and incoherent magnetic fluctuations above T_N . Below T_N , however, the exchange field mixes the individual j_{eff} levels, opening additional channels for exciting the single-ion ground state to spin-orbit entangled excited states, and propagates them across neighbor Co sites. As both the mixing between j_{eff} levels and the coupling of localized spin-orbit excitons to collective magnetic modes increase at low temperature (Refs. 58 and 78 in the revised manuscript), the conversion and transmission of angular momentum through CoO are expected to increase, leading to the enhanced torque efficiency below T_N .

Figure R2. Detail of the torque efficiency in Fig. 4f in the temperature range 100 - 300 K.

In agreement with this scenario, the enhancement of the orbital torque below the Néel temperature is consistently observed in samples of CuOx/CoO/Co with different CoO thickness (see Fig. 4f, Figs. S9 and S10 as well as Fig. R3 below), but is absent in

MnO where magnetic excitations have a pure spin character (Fig. 5 in the revised manuscript as well as Fig. R1).

We have modified the discussion of our results to address this comment, following the arguments presented above.

(4) In order to verify the contribution of the CoO magnons at low temperature, the CoO thickness dependence of the orbital torque efficiency must be determined. The CoO thickness dependence is not a simple exponential decay if contributed by the coherent AFM magnons (Roman Khymyn et al., Phys. Rev. B 93, 224421 (2016)).

In the theoretical work by Khymin et al. (Ref. 53), the spin current is carried by the excitation of a pair of evanescent spin wave modes, with frequencies that are much lower than the antiferromagnetic resonance. The decay of the spin current inside the antiferromagnet can be nonmonotonic or resemble an exponential function depending on the phase shift between the two modes. Therefore, a fit of the torque efficiency vs thickness of CoO will not uniquely determine whether evanescent spin wave modes are responsible for the angular momentum transfer. Moreover, the excitation and population of magnons will vary with CoO thickness, as different thicknesses correspond to different Néel temperatures. Considering these factors, the interpretation of the thickness dependence becomes nontrivial.

In the first version of the manuscript, we reported the dependence of the torque efficiency on CoO thickness measured at room temperature, which showed a monotonic decay with increasing CoO thickness (Fig. 3c). Following the suggestion of the referee, we conducted temperature-dependent torque measurements on newly fabricated Cu^{*}(7)/CoO(t)/Co(3) samples (second batch, newly fabricated samples). As shown in Fig. R3 a, the orbital torque increases below the blocking temperature in all these samples, with thicker CoO layers resulting in smaller torques. The dependence of blocking temperature (T_B) on CoO thickness is presented in Fig. R3b, which is obtained from the exchange bias measurement after field cooling (0.3 T). Additionally, the CoO thickness dependence of the orbital torque at 300 K and 50 K can be found in Figs. R3 c and d, respectively. These additional measurements confirm that the temperature trends reported for CuOx(7)/CoO(2)/Co(3) in Fig. 4f are shared by samples with different CoO thickness. In particular, the torque efficiency first decreases (in absolute value) upon approaching the Néel temperature from above, and then increases strongly at low temperature. The decay constant of the torque efficiency in this new batch of samples is approximately 2.3 nm (3.4 nm) at room temperature (50 K).

This behaviour is consistent with the interpretation of the data presented in response to points 2 and 3 above, based on the local excitation of spin-orbit excitons at the CuOx/CoO interface and their propagation by thermal fluctuations ($T > T_N$) and coherent Néel dynamics ($T < T_N$). In this scenario, the onset of antiferromagnetic order increases both the efficiency of the orbital-to-spin conversion process— by inducing the

admixture of different spin-orbit split crystal field manifolds—and the propagation of angular momentum across CoO by long-range magnetic exchange interactions.

The data in Fig. R3 also confirm that samples with a reduced thickness of CoO (1 nm) present a significantly smaller torque efficiency compared to samples with larger thickness, showing that the formation of a continuous CoO layer with robust antiferromagnetic order is key to the conversion and transmission of angular momentum.

We have included the data from this second batch of samples in the supplementary materials and mentioned this point in the discussion.

Figure R3. **a**, Temperature dependence of the orbital torque efficiency in newly fabricated Cu*(7)/CoO(t)/Co(3) samples. **b**, CoO thickness dependence of the blocking temperature (T_B) from hysteresis loop measurements. **c**, Orbital torque efficiency as a function of CoO thickness at 300 K (above the blocking temperature). **d**, Orbital torque efficiency as a function of CoO thickness at 50 K (below the blocking temperature).

(5) It should be emphasized that the transmission of orbital angular momentum in an AFM insulator is very challenging (Randy S. Fishman et al., Phys. Rev. Lett. 129, 167202 (2022)). In proposing the spin-orbital mixing conductance across the CoO layer, the authors must provide a clear interpretation of why the transmission of orbital angular momentum in an insulating CoO is possible.

The paper by Fishman et al. refers to the orbital angular momentum (OAM) of spin wave fields with spatially twisted phase structure. They consider OAM-carrying magnons originating from tailored exchange geometries (zigzag, honeycomb) and avoided crossings of magnon bands in k -space in the absence of spin-orbit coupling. We believe that this type of magnons is not relevant to the discussion of our results.

Regardless of the phase and twisting of the magnon wavefronts, when the ionic orbital angular momentum is not quenched, magnonic excitations propagating via exchange necessarily involve also the orbital degree of freedom. In other words, magnons propagating in materials like CoO carry both a spin and an orbital component, as part of the coupled angular momentum j .

The mixed spin-orbital character of magnons (or magnetic excitons – as sometime referred to in the old literature) is well-known in antiferromagnets including Co^{2+} ions (see our reply to point 2, and Refs. 57,59,80 and 82 in the revised manuscript). However, with few exceptions (see, e.g., Phys. Rev. Lett. 125, 117209 (2020)), the vast majority of recent works on magnonic and spintronic systems considers magnons as spin-only excitations, which is appropriate in systems with small spin-orbit coupling and nearly quenched orbital magnetic moments. This is clearly not the case of CoO.

An alternative approach sometimes adopted for Kramers' ions with finite orbital momentum consists in projecting the Heisenberg Hamiltonian for total angular momentum onto the lowest Kramers' doublet and derive the magnon dispersion using conventional spin wave theory. This approach, however, only works if the Kramers' doublet is well-separated in energy from all the other excited states. This is not the case for CoO, where the spin-orbit coupling, antiferromagnetic exchange, and crystal field distortion all have comparable energies of order 10 meV.

The discussion of spin-orbital mixing conductance has been revised in the manuscript to include these considerations and those reported in points 2-5 above.

(6) In order to incorporate the spin-flop coupling and the exchange spring effect into the interpretation, it is necessary to measure the coupling at the CoO/Co interface. For instance, spin Hall magnetoresistance (Dazhi Hou et al., Phys. Rev. Lett. 118, 147202 (2017)).

We thank the referee for pointing out this auxiliary control experiment. The spin-flop coupling at the CoO/Co interface has been well studied (J. Appl. Phys. 83, 7219–7221 (1998)), where the perpendicular component (Néel vector perpendicular to the applied field) coexists with the parallel component (parallel to exchange bias field). When spin-flop coupling is dominant, a sign change in the spin Hall magnetoresistance (SMR) is typically expected (Phys. Rev. Lett. 118, 147202 (2017), now included as Ref. 46 in the revised manuscript).

In our previous discussion in the manuscript, we concluded that spin-flop coupling and the exchange spring effect are unlikely to contribute to the orbital torque in $\text{Cu}^*/\text{CoO}/\text{Co}$, as the torque efficiency does not depend on the field-cooling direction. Following the referee's advice, we performed SMR-like measurements (orbital Rashba-Edelstein magnetoresistance in our case) on $\text{Cu}^*(7)/\text{CoO}(2)/\text{Co}(3)$ at both 50 K and 300 K, and did not observe any sign change in the magnetoresistance signal. We therefore conclude that spin flop coupling cannot explain the increase of the torque observed at low temperature. This result indirectly support our proposed explanation

of orbital torque mediated by efficient excitation and propagation of spin-orbit excitons in the antiferromagnetically ordered state.

Figure R4. Orbital Rashba-Edelstein magnetoresistance measurements of Cu*(7)/CoO(2)/Co(3) at 50 K and 300 K. The angle ϕ represents the orientation of the applied field relative to the current direction. The magnitude of the applied field is 0.94 T. The field is not large enough to saturate the magnetization of the Co layer, thus the curve slightly deviates from the usual $\cos^2(\phi)$ dependence.

(7) The comparison of Cu*/CoO/Co, Cu*/Co and SiN/Cu/CoO/Co is not quite convincing. The interfaces are not adequately controlled in this comparison. To substantiate the assertion that the sign change in the orbital torque is attributable to an inverted oxygen gradient, more direct evidence is necessary.

The assignment of the sign change of the orbital torque in Cu*/CoO/Co relative to Cu*/Co to the inverted oxygen gradient is not based on the simple comparison of the nominally different samples, but on extensive and quantitative XPS depth-profiling, as summarized in Fig. 1 and detailed in the Supplementary Material. While we agree with the referee that the quality of the interfaces in sputtered systems cannot always be controlled with high accuracy, we believe that the characterization of the stoichiometry profile and roughness reported in the manuscript supports our conclusions. Moreover, we provide additional evidence by comparing the torques in Cu*/CoO/Co and SiN/Cu/CoO/Co, which exhibit the same negative sign, indicating that the top oxidation of Cu does not affect the sign of the torque. As further comparison, we find no sizable torque in CoO/Co, leading us to conclude that the negative torque is related to the Cu/CoO interface. In this resubmission, we also demonstrate that the negative torque can be achieved by replacing CoO with NiO, which provides strong evidence that the Cu/oxide interface is responsible for the observed negative torque.

We note that the orbital Rashba-Edelstein effect is expected in the presence of an oxygen gradient (Phys. Rev. B 103 (12), L121113 (2021), Ref. 30 in the revised manuscript), and other studies indicate that the Cu/oxide interface is sufficient to generate an orbital current (Phys. Rev. B 103 (2), L020407 (2021), Ref. 4 in the revised manuscript). Therefore, we attribute the negative torque to the inverted oxygen gradient at the Cu/CoO interface.

To further clarify this point, we have included torque data from Cu*/NiO/Co in the Fig. 5 for revised manuscript, highlighting the importance of the Cu/oxide interface in the generation of the negative torque.

References cited in this reply with hyperlinks:

2. Go, D. et al. Long-Range Orbital Torque by Momentum-Space Hotspots. Phys. Rev. Lett. 130, 246701 (2023).
4. Kim, J. et al. Nontrivial torque generation by orbital angular momentum injection in ferromagnetic-metal/Cu/Al₂O₃ trilayers. Phys. Rev. B 103, L020407 (2021).
30. Go, D. et al. Orbital Rashba effect in a surface-oxidized Cu film. Phys. Rev. B 103, L121113 (2021).
36. Hahn, C. et al. Conduction of spin currents through insulating antiferromagnetic oxides. Europhys. Lett. 108, 57005 (2014).
37. Wang, H., Du, C., Hammel, P. C. & Yang, F. Antiferromagnonic Spin Transport from Y₃Fe₅O₁₂ into NiO. Phys. Rev. Lett. 113, 097202 (2014).
38. Wang, H., Du, C., Hammel, P. C. & Yang, F. Spin transport in antiferromagnetic insulators mediated by magnetic correlations. Phys. Rev. B 91, 220410 (2015).
39. Qiu, Z. et al. Spin-current probe for phase transition in an insulator. Nat. Commun. 7, 12670 (2016).
40. Li, Q. et al. Coherent ac spin current transmission across an antiferromagnetic CoO insulator. Nat. Commun. 10, 5265 (2019).
41. Dabrowski, M. et al. Coherent Transfer of Spin Angular Momentum by Evanescent Spin Waves within Antiferromagnetic NiO. Phys. Rev. Lett. 124, 217201 (2020).
42. Lin, W., Chen, K., Zhang, S. & Chien, C. L. Enhancement of Thermally Injected Spin Current through an Antiferromagnetic Insulator. Phys. Rev. Lett. 116, 186601 (2016).
43. Prakash, A., Brangham, J., Yang, F. & Heremans, J. P. Spin Seebeck effect through antiferromagnetic NiO. Phys. Rev. B 94, 14427 (2016).
44. Baldrati, L. et al. Spin transport in multilayer systems with fully epitaxial NiO thin films. Phys. Rev. B 98, 014409 (2018).
45. Hoogeboom, G. R. et al. Role of NiO in the nonlocal spin transport through thin NiO films on Y₃Fe₅O₁₂. Phys. Rev. B 103, 144406 (2021).
46. Hou, D. et al. Tunable Sign Change of Spin Hall Magnetoresistance in Pt/NiO/YIG Structures. Phys. Rev. Lett. 118, 147202 (2017).
47. Hasegawa, K., Hibino, Y., Suzuki, M., Koyama, T. & Chiba, D. Enhancement of spin-orbit torque by inserting CoOx layer into Co/Pt interface. Phys. Rev. B 98, 020405 (2018).
48. Zhu, L., Zhu, L. & Buhrman, R. A. Fully Spin-Transparent Magnetic Interfaces Enabled by the Insertion of a Thin Paramagnetic NiO Layer. Phys. Rev. Lett. 126, 107204 (2021).

49. Lin, W. & Chien, C. L. Electrical Detection of Spin Backflow from an Antiferromagnetic Insulator Y₃Fe₅O₁₂ Interface. Phys. Rev. Lett. 118, 067202 (2017).
50. Rezende, S. M., Rodríguez-Suárez, R. L. & Azevedo, A. Diffusive magnonic spin transport in antiferromagnetic insulators. Phys. Rev. B 93, 054412 (2016).
51. Morita, T., Koyama, T. & Chiba, D. Process gas dependence of spin-orbit torque in Pt/NiO/Co structures. Appl. Phys. Lett. 122, 262402 (2023).
52. Takei, S., Moriyama, T., Ono, T. & Tserkovnyak, Y. Antiferromagnet-mediated spin transfer between a metal and a ferromagnet. Phys. Rev. B 92, 020409 (2015).
53. Khymyn, R., Lisenkov, I., Tiberkevich, V. S., Slavin, A. N. & Ivanov, B. A. Transformation of spin current by antiferromagnetic insulators. Phys. Rev. B 93, 224421 (2016).
54. Wang, Y. et al. Magnetization switching by magnon-mediated spin torque through an antiferromagnetic insulator. Science 366, 1125–1128 (2019).
55. Zhu, D. et al. Sign Change of Spin-Orbit Torque in Pt/NiO/CoFeB Structures. Phys. Rev. Lett. 128, 217702 (2022).
57. Sakurai, J., Buyers, W. J. L., Cowley, R. A. & Dolling, G. Crystal Dynamics and Magnetic Excitations in Cobaltous Oxide. Phys. Rev. 167, 510–518 (1968).
58. Keisuke, T. & Shinichi, I. Magnetic Excitations in CoO. J. Phys. Soc. Japan 75, 084708 (2006).
59. Sarte, P. M. et al. Spin-orbit excitons in CoO. Phys. Rev. B 100, 075143 (2019).
74. Kanamori, J. Theory of the Magnetic Properties of Ferrous and Cobaltous Oxides, I. Prog. Theor. Phys. 17, 177–196 (1957).
75. Kanamori, J. Theory of the Magnetic Properties of Ferrous and Cobaltous Oxides, II. Prog. Theor. Phys. 17, 197–222 (1957).
76. van Schooneveld, M. M. et al. Electronic Structure of CoO Nanocrystals and a Single Crystal Probed by Resonant X-ray Emission Spectroscopy. J. Phys. Chem. C 116, 15218–15230 (2012).
77. Schrön, A., Rödl, C. & Bechstedt, F. Crystalline and magnetic anisotropy of the 3d-transition metal monoxides MnO, FeO, CoO, and NiO. Phys. Rev. B 86, 115134 (2012).
78. Chou, H. & Fan, H. Y. Light scattering by magnons in CoO, MnO, and α -MnS. Phys. Rev. B 13, 3924–3938 (1976).
79. Kant, C. et al. Optical spectroscopy in CoO: Phononic, electric, and magnetic excitation spectrum within the charge-transfer gap. Phys. Rev. B 78, 245103 (2008).
80. Sarte, P. M., Wilson, S. D., Attfield, J. P. & Stock, C. Magnetic fluctuations and the spin-orbit interaction in Mott insulating CoO. J. Phys. Condens. Matter 32, 374011 (2020).
81. Kim, J. et al. Magnetic Excitation Spectra of Sr₂IrO₄ Probed by Resonant Inelastic X-Ray Scattering: Establishing Links to Cuprate Superconductors. Phys. Rev. Lett. 108, 177003 (2012).
82. Daniel, M. R. & Cracknell, A. P. Magnetic Symmetry and Antiferromagnetic Resonance in CoO. Phys. Rev. 177, 932–941 (1969).

Reviewer #2 (Remarks to the Author):

The generation, transmission, and conversion of orbital torque are very important for the electrical control of magnetization in nanoscale devices. In this paper, the authors demonstrate a large orbital torque and enhanced orbital-to-spin conversion by an antiferromagnetic insulating CoO layer. They found that inserting CoO in CuOx/CoO/Co trilayers reverses the sign of torque and triples its magnitude compared to the CuOx/Co, which is explained by the inverted oxygen gradient at the CuOx/CoO as compared with the Cu/Co interface. Moreover, at low temperatures, the onset of antiferromagnetic order and exchange bias induces a further many-fold increase of the torque. These results reveal that antiferromagnetic insulators like CoO can provide highly effective orbital-to-spin transduction to improve the performance of spintronic devices. The experimental results are very interesting, and the analysis on the experimental results is very reasonable. However, some issues should be clarified before its publication in NC.

We thank the referee for the critical reviewing of our work, pointing out its strong interest and solid analysis of the experimental results. We have taken into account the referee's criticism to improve the clarity of the manuscript, as detailed below.

1. In 102-104 lines: Elemental depth profiling by X-ray photoelectron spectroscopy (XPS) shows that natural oxidation of Cu leads to the formation of Cu₂O and an oxygen gradient extending over about 2 nm from the top surface, see Fig. 1c. In 385-386 lines: The samples with Cu were naturally oxidized in the air for two days after fabrication, the 3 nm CuOx on top prevents the further oxidization of the stacks. "An oxygen gradient extending over about 2 nm from the top surface" may be in conflict with "the 3 nm CuOx on top".

We apologize for the confusion. The reported thickness of CuOx refers to the total thickness of the Cu layer prior to oxidation. Following oxidation, the XPS results indicate an oxygen penetration length of approximately 2 nm, suggesting that depositing a Cu layer thicker than 2 nm results in approximately 2-nm-thick oxidized Cu and 1-nm-thick Cu metal. Protecting the underlying Cu metal from oxidation is crucial for generating an orbital current, as fully oxidized CuOx layers are found to generate a negligible orbital torque, in line with theoretical works that indicate the oxidation gradient in Cu as the source of the orbital current. Thus, starting from deposited Cu layers with thickness < 2 nm, the orbital torque first increases with Cu thickness. However, increasing the total Cu thickness beyond 7 nm, the orbital torque reduces due to significant shunting of the charge current through metallic Cu, which is not a good generator of orbital current.

We have clarified this point in the revised manuscript.

2. In 116-118 lines, Fig.1 c shows the relative concentration of elemental and oxide

chemical species vs depth profiles determined by XPS combined with Ar⁺ etching of Cu^{*}(7)/Co(5), Cu^{*}(1.5)/Co(5), and Cu^{*}(7)/CoO(2)/Co(5). The peak positions of Cu and Co in Fig. 1 c seem contradictory to the thickness of the Cu and Co layers. For example, in the bottom panel of Fig. 1c, the Co peak position of Cu^{*}(7)/CoO(2)/Co(5) is located at about 6.5 nm. How thick is the detecting depth of the XPS?

We thank the referee for the critical question regarding the XPS data. The depth resolution of our XPS measurements is approximately 1 nm. This results in the broadening of the XPS peaks, as observed in Figure 1c. The element-specific depth profiles are obtained by combining XPS with Ar⁺ etching of the thin film, with the sample thicknesses shown in Figure 1c. The etching rate was calibrated to 1.1 nm/min on SiO₂ on Si, which is assumed to be constant across different metals for the thickness calculation in Figure 1c. However, since the etching rate may vary for different metals and compounds, the thickness values in Figure 1c may differ slightly from the actual values.

It is noteworthy that in XPS depth profiling, the layer thickness should be calculated as the difference between the two inflection points and not in correspondence of the gaussian maximum. In the case of the Cu^{*}(7)/CoO(2)/Co(5) sample (bottom panel of figure 1c), the thickness of the Co layer is estimated to be 8.5 nm – 4 nm = 4.5 nm, considering the interfaces Co/Si and CoO/Co. We have clarified how the layer thickness is calculated from XPS depth profiling in the Methods section.

The main conclusions of our paper remain unaffected, as the oxygen gradient and metal/oxide interface are confirmed through the depth-dependent XPS measurements.

3. In 34-36 Lines, the authors mentioned that similar to spin torques, orbital torques can be used for magnetization switching, domain wall motion, and excitation of high-frequency dynamics in nano-oscillators. One wonders whether the enhanced orbital torques can switch the magnetization of Co layer in their samples. If not, please give a brief discussion on it.

Indeed, the orbital torque exhibits the same symmetry as the spin torque and can be used for magnetization switching and domain wall motion, as shown by recent studies (e.g., Commun. Phys. 4, 234 (2021); Nano Lett. 23, 11323–11329 (2023); arXiv:2412.04872 (2024); arXiv:2410.02238 (2024); Nat. Commun. 15, 8645 (2024)). Our samples have easy plane magnetic anisotropy. This property makes them ideal for studying the torque efficiency, but less ideal for switching by either an orbital torque or spin torque because the in-plane magnetization lacks bistability. Ideally, a switching device would incorporate a magnetic thin film with perpendicular magnetic anisotropy. This can be achieved in CoO/Co thin films with a Co/AlO_x or Co/MgO interface (see, e.g., Feng et al., Phys. Rev. Appl. 13, 044029 (2020)). However, such a study is beyond the scope of the present work. We have added a note to the manuscript to

comment on this point, indicating the interest of using CoO as efficient spin-orbital conversion layer for magnetization switching by orbital torques.

Reviewer #3 (Remarks to the Author):

In this work, the authors showed a notable orbital torque in CuOx/CoO/Co trilayer, where the orbital current generated by CuOx passed through and converted to spin current in CoO layer, resulting in a sizable torque efficiency. This work is interesting, and may open a possibility that using insulating antiferromagnets as the spacers for orbital-to-spin conversion. I can recommend the publication if the concerns below are addressed:

We thank the referee for their insightful comments and for recognizing the interest and novelty of our work concerning insulating antiferromagnets as efficient orbital-to-spin converters.

1. The authors suggested that CoO layer is insulating. They should provide evidence that the thin CoO layer is indeed insulating, otherwise CoO layer might be just like an ordinary metallic orbital-to-spin converter.

This is a critical point, which we examined by direct measurement of the electrical resistance as well as by comparing the results obtained in different samples and as a function of temperature:

The four-point resistance of 2-nm-thick and 27-nm-thick single CoO layers is found to exceed the measurement range of the multimeter (200 M Ω), confirming that our sputtered CoO layers have good insulating properties ($\rho > 182 \Omega \text{ cm}$).

We compared the longitudinal resistance of Cu*(7)/CoO(t)/Co(3) in samples with different CoO thickness. As shown in Figure R5, we observe no significant change in resistance as the CoO thickness varies between 1 and 10 nm, indicating that the resistance of the CoO layer is significantly larger than that of the metallic Cu and Co layers for all samples investigated in this work.

Figure R5. Dependence of longitudinal resistance in Cu*(7)/CoO(t_{CoO})/Co(3) as a function of CoO thickness.

The fact that the torque efficiency increases at low temperature further excludes that the orbital-to-spin conversion and transmission are mediated by electron hopping, as the conductivity of CoO is known to reduce strongly with decreasing temperature (see,

e.g., Kant et al., PRB 78, 245103 (2008)). We have added the above information to the Supplementary Material (new Section S3).

2. The authors showed that field cooling along different directions does not influence the results. However, the field applied is too small, which might not be sufficient to influence the AFM state. I suggest the authors check the results using a much larger cooling field.

This comment pertains to the effects of the cooling field on the exchange bias effect. The Néel temperature of the 2 nm CoO layer is approximately 200 K, and at 300 K, CoO is in the paramagnetic phase. The coercivity of the ferromagnetic Co layer at 300 K is less than 20 mT, meaning the 0.3 T cooling field is more than sufficient to saturate all the magnetic moments of Co in the field direction. As the temperature decreases in the presence of the 0.3 T cooling field, dropping below the Néel temperature of CoO, the Néel vector of CoO aligns with the magnetization direction of Co due to the exchange interaction at the interface. The interfacial exchange field, which is much larger than the external field, determines the exchange bias effect.

In exchange bias systems involving ferromagnet/antiferromagnet bilayers (excluding noncollinear antiferromagnets), the exchange bias field does not significantly depend on the cooling field as long as the field exceeds the coercivity of the ferromagnet (JMMM 192, 203-232 (1999); Phys. Rev. B 94, 134422 (2016)). Therefore, we argue that the 0.3 T field applied during field cooling is sufficient to saturate the Co magnetization and align the Néel order of CoO layer below its Néel temperature.

3. Since there is an exchange bias, it is possible to realize an efficient field free switching of perpendicular magnetization. It would be good to show this in experiment, or at least add some discussion to mention this possibility.

We thank the referee for this comment, which is similar to a point raised by Referee 2. Our work demonstrates a large torque efficiency in CuOx/CoO/Co, which, in principle, could lower the critical switching current by using current-induced torque, as the critical current is inversely proportional to the torque efficiency. However, achieving deterministic switching in easy-plane magnetic layers is not straightforward.

As the referee pointed out, achieving field-free switching in a perpendicular magnetized film using the exchange bias effect is indeed possible. This makes CoO coupled to a source of orbital current also an interesting system also for practical applications. However, orbital torque switching is outside the scope of our current study and would require specific engineering of the Co magnetic anisotropy. In the revised manuscript, we nonetheless discuss have discussed the potential for high-efficiency, field-free switching with orbital torque from CuOx/CoO.

SUMMARY OF CHANGES

- We report measurements on new samples where CoO has been replaced by NiO and MnO in order to probe how the orbital torque efficiency depends on the orbital magnetic moment and spin-orbit split low-energy multiplet structure of the antiferromagnetic oxides. The measurements of these samples have been included in Fig. 5 and Fig. S12.
- We added measurements of the electrical resistance of Cu*/CoO(t)/Co layers as a function of CoO thickness (Fig. S3).
- We report measurements performed on a new batch of Cu*(7)/CoO(t)/Co(3) samples to probe the thickness and temperature dependence of the orbital torque efficiency as well as the dependence of blocking temperature on CoO thickness. The measurements of these samples have been included in Fig. S10.
- Added measurements of the orbital Rashba-Edelstein magnetoresistance of Cu*(7)/CoO(2)/Co(3) at 300 K and 50 K that indicate the absence of spin-flop coupling in this temperature range (Fig. S11).

In addition to the above, we have substantially revised the presentation and discussion of the data in the main text to take into account the comments and queries by the referees. In particular, we included:

- Extended revised discussion of the mechanisms underpinning the conversion and transport of orbital momentum by CoO. We now compare different theoretical models (incoherent and coherent magnons, spin-flop coupling) and propose the spin-orbit exciton model as the most likely explanation of the enhanced orbital torque efficiency induced by antiferromagnetic CoO.
- Revised discussion of the sign of the negative torque.
- Information on sample thickness, clarifying the thickness of oxidized and metallic Cu.
- Outlook on reducing critical switching current in perpendicular magnetized thin film coupled to CoO and an orbital current source.
- Revised/new Sections S3, S9-S12 in the Supplementary Material
- Additional references in main text (1-16 reported below)

All changes to the text have been highlighted in blue in the revised manuscript and Supplementary Material.

1. Kanamori, J. Theory of the Magnetic Properties of Ferrous and Cobaltous Oxides, I. Prog. Theor. Phys. 17, 177–196 (1957).
2. Kanamori, J. Theory of the Magnetic Properties of Ferrous and Cobaltous Oxides, II. Prog. Theor. Phys. 17, 197–222 (1957).
3. van Schooneveld, M. M. et al. Electronic Structure of CoO Nanocrystals and a Single Crystal Probed by Resonant X-ray Emission Spectroscopy. J. Phys. Chem. C 116, 15218–15230 (2012).
4. Schrön, A., Rödl, C. & Bechstedt, F. Crystalline and magnetic anisotropy of the 3d-transition metal monoxides MnO, FeO, CoO, and NiO. Phys. Rev. B 86, 115134 (2012).
5. Chou, H. & Fan, H. Y. Light scattering by magnons in CoO, MnO, and α -MnS. Phys. Rev. B 13, 3924–3938 (1976).
6. Kant, C. et al. Optical spectroscopy in CoO: Phononic, electric, and magnetic excitation spectrum within the charge-transfer gap. Phys. Rev. B 78, 245103 (2008).
7. Sarte, P. M., Wilson, S. D., Atfield, J. P. & Stock, C. Magnetic fluctuations and the spin–orbit interaction in Mott insulating CoO. J. Phys. Condens. Matter 32, 374011 (2020).
8. Kim, J. et al. Magnetic Excitation Spectra of Sr₂IrO₄ Probed by Resonant Inelastic X-Ray Scattering: Establishing Links to Cuprate Superconductors. Phys. Rev. Lett. 108, 177003 (2012).
9. Daniel, M. R. & Cracknell, A. P. Magnetic Symmetry and Antiferromagnetic Resonance in CoO. Phys. Rev. 177, 932–941 (1969).
10. Csiszar, S. I. et al. Controlling Orbital Moment and Spin Orientation in CoO Layers by Strain. Phys. Rev. Lett. 95, 187205 (2005).
11. Neubeck, W. et al. Orbital moment determination of simple transition metal oxides using magnetic X-ray diffraction. J. Phys. Chem. Solids 62, 2173–2180 (2001).
12. Flipse, C. F. J., Rouwelaar, C. B. & Groot, de, F. M. F. Magnetic properties of CoO nanoparticles. Eur. Phys. J. D 9, 479–481 (1999).
13. Fernandez, V., Vettier, C., de Bergevin, F., Giles, C. & Neubeck, W. Observation of orbital moment in NiO. Phys. Rev. B 57, 7870–7876 (1998).
14. Gupta, R. et al. Harnessing orbital Hall effect in spin-orbit torque MRAM. Nat. Commun. 16, 130 (2025).
15. Fukami, S., Zhang, C., DuttaGupta, S., Kurenkov, A. & Ohno, H. Magnetization switching by spin–orbit torque in an antiferromagnet–ferromagnet bilayer system. Nat. Mater. 15, 535–541 (2016).
16. van den Brink, A. et al. Field-free magnetization reversal by spin-Hall effect and exchange bias. Nat. Commun. 7, 10854 (2016).